

# Opposite asymmetries of face and trunk and of kissing and hugging, as predicted by the axial twist hypothesis

Marc H.E. de Lussanet

Department of Movement Science, University of Münster, Münster, Germany
Otto Creutzfeldt Center for Cognitive and Behavioral Neuroscience, University of Münster, Münster, Germany

## ABSTRACT

The contralateral organization of the forebrain and the crossing of the optic nerves in the optic chiasm represent a long-standing conundrum. According to the Axial Twist Hypothesis (ATH) the rostral head and the rest of the body are twisted with respect to each other to form a left-handed half turn. This twist is the result, mainly, of asymmetric, twisted growth in the early embryo. Evolutionary selection tends to restore bilateral symmetry. Since selective pressure will decrease as the organism approaches symmetry, we expected a small control error in the form of a small, residual right-handed twist. We found that the mouth-eyes-nose (rostral head) region shows a left-offset with respect to the ears (posterior head) by up to 0.8° ($P < 0.01$, Bonferroni-corrected). Moreover, this systematic aurofacial asymmetry was larger in young children (on average up to 3°) and reduced with age. Finally, we predicted and found a right-sided bias for hugging (78%) and a left-sided bias for kissing (69%). Thus, all predictions were confirmed by the data. These results are all in support of the ATH, whereas the pattern of results is not (or only partly) explained by existing alternative theories. As of the present results, the ATH is the first theory for the contralateral forebrain and the optic chiasm whose predictions have been tested empirically. We conclude that humans (and all other vertebrates) are fundamentally asymmetric, both in their anatomy and their behavior. This supports the thesis that the approximate bilateral symmetry of vertebrates is a secondary feature, despite their being bilaterians.

## INTRODUCTION

A well-known mystery of the forebrain is its contralateral organization, which means that the left side of the cerebrum and thalamus connect predominantly to the right side of more caudal regions of the central nervous system (CNS) and vice versa. This peculiar contralateral organization is present in every vertebrate known, even agnathans, but in no other (invertebrate) animal. The same is true for the optic chiasm, which projects the right visual field to the left optic tectum of the midbrain (and, in tetrapods and bony fishes, also to the left visual cortex) and vice versa.

Corresponding author
Marc H.E. de Lussanet,
lussanet@uni-muenster.de

For over a century, various theories have been proposed to explain this contralateral organization (and sometimes also the optic chiasm), but all have been invalidated on the basis of inconsistencies (*Vulliemoz, Raineteau & Jabaudon, 2005*; *De Lussanet & Osse, 2012*). For example, the still popular visual map theory by Cajal proposes that the function of the optic chiasm is to draw a continuous visual map in the brain (*Ramón y Cajal, 1899*; *Loosemore, 2009*). However, due to the decussation of the optic radiation, the visual cortex does not draw such a continuous visual map (*De Lussanet & Osse, 2015*; *Nieuwenhuys, Voogd & van Huijzen, 2008*).

According to a very different theory, the contralateral organization did not evolve as a feature of the brain, but rather is the result of a half turn of the rostral part of the head and the body (starting from the posterior head including the gill/ear region) with respect to each other. This idea of a 180-degree twist was first expressed, but not worked out, in an early abstract (*Kinsbourne, 1978*). More recently, the idea was proposed again independently, and worked out into a developmental and evolutionary model, christened as the Axial Twist Hypothesis (ATH) (*De Lussanet & Osse, 2012*; *De Lussanet & Osse, 2015*).

According to the ATH, the twist between the rostral head and the rest of the body develops by oppositely directed asymmetric, twisted growth. It is generally accepted that bilateral symmetry, as widely found in bilaterian animals, is not a feature that arises automatically. Rather, bilateral symmetry represents an evolutionary advantage, for example in locomotion (*Beklemishev, 1969*) and in sexual selection (*Grammer & Thornhill, 1994*). Bilateral symmetry is never perfect, due to so-called developmental accidents (*Van Valen, 1962*). Consequently, bilateral symmetry will show a distribution in any natural population. In addition, the symmetry breakage can have a systematic directional bias (*Hallgrímsson et al., 2005*; *Steele, 2002*). As is typical for control systems, the extent of evolutionary pressure that drives developmental processes depends on the degree of deviation from optimality. As the system approaches the optimum, evolutionary pressure ceases so that finally, a small residual control error is expected to remain at the population level.

Since the evolutionary selection for bilateral symmetry is due to locomotion (*Galis et al., 2014*; *Beklemishev, 1969*) and sexual selection (*Møller & Thornhill, 1998*; *Graham & Özener, 2016*; *Little et al., 2008*), it will only affect the external shape of the body (for example, in flatfishes the eyes and mouth migrate, but not the inner ear and the brain; cf. *Leech, 1923*). Internal asymmetries, such as the orientation and location of heart, lungs, liver, stomach, and bowels are therefore predicted to be evolutionary neutral and so to reflect the history of vertebrate evolution. Thus, these structures are not involved in the axial twist but instead retain their original asymmetric orientation.

Moreover, the direction of asymmetric growth (*De Lussanet & Osse, 2012*) follows from (1) the direction of the whirl in the early primitive streak in bird embryos (*Lopez-Sanchez, Garcia-Martinez & Schoenwolf, 2001*; *Kirby et al., 2003*), (2) the relative turning of the heart and the soma in embryonic development, (3) the direction of asymmetric growth of the eyes and the posterior head region in zebrafish (*Keller et al., 2008*; *De Lussanet & Osse, 2012*) and (4) the left-eye optic tract passing above the right-eye optic tract in the optic chiasm in those animals in which the tracts do not merge (e.g., teleost fish, catfish,
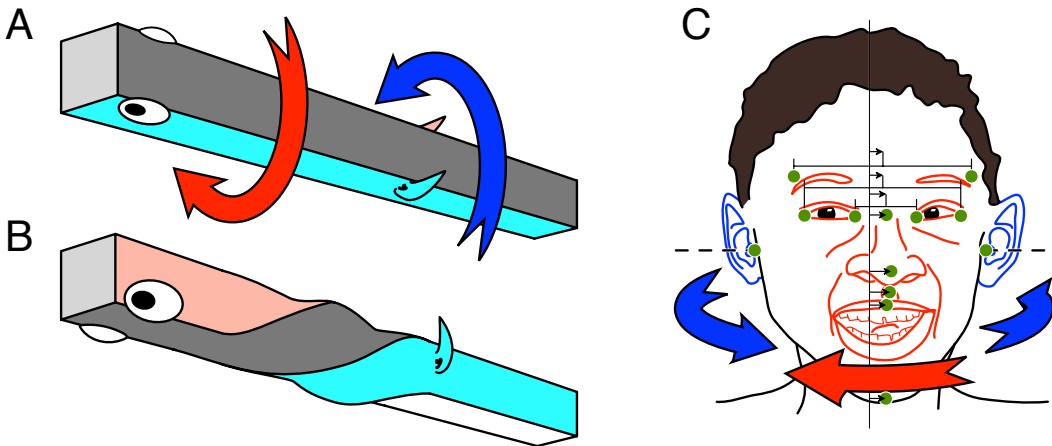

**Figure 1** **(A–B) Schema of the asymmetric development leading to the left-handed twist. (C) Schema of the predicted aurofacial asymmetry for the human face.** (A) The early embryo is at first positioned on its left side. The facial region, including the forebrain, grows asymmetrically (red arrow) to become symmetric. (B) The ears (gill region) with the rostral head and the soma reach symmetry by opposite asymmetric growth (blue arrow). (C) The eyes-nose-mouth region (red) belongs anatomically to the rostral head, whereas the ears (blue) are part of the posterior head. Since the asymmetric growth (red and blue arrows) is predicted to be incomplete, we hypothesize that the face is shifted to the left side (small arrows) with respect to the mid-plane between the tragi (vertical line). Green dots show paired and median facial landmarks; paired: tragus of the ear, frontotemporale, exo- and endocanthion of the eye; median from top: nasion, pronasale, subnasale, labiale superior, and menton. Small arrows show the asymmetry of the facial landmarks.

chameleon: *Polyak, 1957*; *Nieuwenhuys, Ten Donkelaar & Nicholson, 1998*). Accordingly, the direction of the twist is left-handed (Figs. 1A–1B).

The original idea of the twist (*Kinsbourne, 1978*) was recently published (*Kinsbourne, 2013*) and christened Somatic Twist Model (STM). According to the STM, the body (soma) is twisted by 180° with respect to the rostral head. The STM and the ATH both explain the Dorsoventral Inversion Hypothesis (DIH) (*Geoffroy-Saint-Hilaire, 1822*; *Nübler-Jung & Arendt, 1994*; *Lowe et al., 2003*), provided that the mouth twists with the rostral head (*De Lussanet & Osse, 2015*). The twist of the mouth is still an open question, because the mouth develops very late in vertebrates (*De Lussanet & Osse, 2012*; *De Lussanet & Osse, 2015*). However, on the basis of the possible evolutionary scenarios, it is thought likely that the mouth region twists with the rostral head (*Kinsbourne, 2013*; *De Lussanet & Osse, 2015*).

The ATH thus has considerable explanatory power for a wide range of morphological structures, and is supported by developmental observations. For the present study we planned to validate the hypothesis by generating and testing some new predictions on humans.

In the present study we made, on the basis of the ATH, new predictions for asymmetries of the human face and for interactive social behavior. According to the ATH, the eyes and nose belong to the rostral part of the head, whereas the ears, being evolved from gill structures, belong to the caudal part (Figs. 1A–1B). Therefore, if the twist is incomplete

due to the evolution-based residual control error, the eyes and nose should be positioned slightly to left with respect to the medial plane between the ears. We call this the *Aurofacial Asymmetry* (indicated by the small arrows in Fig. 1C). This is a new kind of facial asymmetry that has never been published.

Since the asymmetry is the result of a developmental process, we predict that maximal symmetry is reached at the end of growth. A major extension of the facial features occurs during adolescent growth, so we hypothesized that the aurofacial asymmetry in preadolescent children is larger than in adult faces.

Social interactive behaviors such as kissing (see, e.g., *Güntürkün, 2003*; *Ocklenburg & Güntürkün, 2009*) and hugging (see, e.g., *Turnbull, Stein & Lucas, 1995*; *Packheiser et al., 2019*) are special in the sense that the two interactants are at first medially aligned but cannot make symmetric medial physical contact. They are forced to aim at either the left or the right side rather than at the midline. In many mammals and birds the left hemisphere controls singing and vocalising and in some (humans and common marmosets) it is known that the right side of the mouth moves more strongly than the left during communicative vocalisations (*MacNeilage, Rogers & Vallortigara, 2009*). The ATH predicts a specific bias for kissing and hugging, but other theories to explain these asymmetries, will be discussed at the end of this paper (see *Ocklenburg et al., 2018* for a recent review).

When kissing, the opposites aim with their mouth. According to the ATH, the mouth has moved medially from the left side of the head's midline, so the region left from the facial midline is, in a somatosensory sense, more associated with the facial centre than the region to the right of the midline. Accordingly, the ATH predicts a slight bias to aim with the left side of the head when kissing. Therefore, when forced to avoid the facial midline (due to the nose that sticks out), the ATH predicts that this bias should lead to a tendency to kiss with the left side of the face. This has the effect that the head is tilted to the right.

In contrast, hugging partners aim for each other with their trunk (rather than their face). Again a medial contact is not possible because the heads are in the way, so that a forced choice to aim with either the left or the right side has to be made by the opposites. Since the trunk belongs to the body region that is caudal with respect to the twist, and thus grows toward the midline from the right side, we predicted an opposite bias for hugging than for kissing.

## METHODS

The current work was approved by the ethics committee of FB07 of the University of Münster, under the reference EK-17-25.

### Analysis of 3D surface data

To analyse the aurofacial asymmetry in adults, we investigated a database that was created on the basis of 200 3D scans of individual human faces (*Troje & Bülthoff, 1996*). The 200 3D surface maps that can be obtained from the database (see http://faces.kyb.tuebingen.mpg.de) represent weighed morphs from the original data, so that none of the faces equal a real life individual. These morphs represent a dense vertex to vertex correspondence map which was originally computed as described in *Blanz &*

*Vetter (1999)*. Correspondence maps were initially obtained with respect to a perfectly symmetric reference face containing $N = 75{,}196$ vertices (the template). Vertices were also arranged symmetrically with half of them in the left and half of them in the right hemiface establishing perfect inter-hemispheric correspondence. Thus, each pair of vertices locates the same anatomical landmark on the left and the right side of each of the 200 faces, as well as across the faces. A pdf of *Blanz & Vetter (1999)*, showing exemplary images of faces, can be downloaded from the website http://faces.kyb.tuebingen.mpg.de. After registration, further (3-D) images are made available at that site.

Each individual face $j$ in the database could thus be described in terms of the locations in 3D space $V_j = \{\vec{v_{i,j}}\}$, $\vec{v_{i,j}} \in \mathbb{R}^3$ for each of the vertices $i = \{1, \ldots, n\}$. The indices of the vertices define the vertex-by-vertex correspondence between each individual face and the bilaterally symmetric template. $C = \{c_i\}$ defines the inter-hemispheric correspondence in the template face, and therefore also for each individual face $j$, with $c_i$ containing the index of the contralateral vertex that corresponds to vertex $i$.

The coordinate system of the template has its origin in the symmetry plane. The frontal plane crosses the tragi of the left and right ears and the orientation of the face about the lateral axis (pitch) is defined by the mean pitch angle across the whole database.

Each vertex in the face was also assigned a horizontal angle $\alpha_i = \arctan(x_i/y_i)$. The average symmetric face, the template, was computed using $\alpha_{symm,i} = (\alpha_{mean,i} - \alpha_{mean,c(i)})/2$. We now quantify the local torsion for vertices $i$ and $c(i)$ as the difference between the angle of each face $j$ with the average symmetric face $\alpha_{torsion,i} = \alpha_i - \alpha_{symm,i}$. A working Matlab® script for calculating the aurofacial asymmetry is provided in FigShare.com (*De Lussanet, 2019*).

## Analysis of 3D landmark data

The aurofacial asymmetry of childrens' faces was analysed on the basis of a different dataset (*Klingenberg et al., 2010*). The 3D locations of thirteen facial landmarks were used for this analysis (bilateral: the tragi of the ears, the exo- and endocanthion of the eyes, and the frontotemporals; medial: the nasion, the pro- and subnasale, the labiale superior, and the menton). We based our analysis on the mean 3D landmark positions as published in *Klingenberg et al. (2010)*. These mean data represented four groups of children, from Finland and from South Africa. There were two age groups, and two experimental groups. The "exposed" children had been diagnosed with fetal alcohol syndrome, but this diagnosis is irrelevant for the purpose of the present study. We took the 3-D coordinates from the frontal, left and right view of the morphed faces for each group. From these average 3D coordinates we calculated the torsion angles for each landmark $\alpha_{torsion,i}$ as described above. The morphs of Klingenberg et al. were exaggerated by a scaling factor $S$, which was $S = 5$ for the young and $S = 10$ for the older age groups respectively. This exaggeration was removed by dividing $\alpha_{torsion,i}/S$.

In order to make these results comparable with the adult data, the same 3D landmarks were selected from the database of 200 young adults (*Blanz & Vetter, 1999*).

## Analysis of frontal photographs

Finally, as an independent measure of the aurofacial asymmetry, two collections of frontal photographs of human faces from a free online database ("nottingham": $N = 439$; "aberdeen": $N = 392$; http://pics.stir.ac.uk) were analysed. All photos were pasted to approximately the same size, using a vector graphics software. The central facial region containing eyes, nose and mouth, was then hidden with an oval to avoid subjective bias of the experimenter. Fifty percent of the faces were mirrored on the basis of a random sequence of plus and minus one. On the basis of the visibility and the position of the ears' tragus and the buccal region, the 113 truly frontal photographs were selected (maximally one per subject). The central ellipses covering the facial region were removed again. Tilted faces were rotated so that the eyes were aligned horizontally. The locations of the tragi and the endocanthion were marked. The faces that had been mirrored were flipped back using the same random sequence and the angle of the lateral shift of the eyes was calculated for each face, as described above. For this, we assumed the endocanthions and the tragus had the same distance to the vertical midline between the tragi (as if the surface of the face corresponded to a vertical cylinder). The analysis and a table of the aurofacial angles is available as Fig. S3.

## Internet search for photographs of kissing and hugging

Our predictions were further tested by analyzing photographic pictures taken from the internet. The most popular search engines gave highly similar results, so we used just one of them. Searches for the same key words on different days did provide partly different results, so the most effective key words were searched twice, on different days.

Most efficient key words for photographs of hugging were: "hugging" (1st $N = 70$, 2nd $N = 128$, 3rd $N = 80$), "omhelzen" (1st $N = 60$, 2nd $N = 76$), "Umarmung" (1st $N = 78$, 2nd $N = 83$), less efficient key words were "hug", "umarmen", "omarmen", "omarming", "embracing", "goodbye airport", "Abschied Bahnhof", "afscheid Schiphol", "wiedersehen". The photos were sorted into the subcategories "with children", "females", "males", and "mixed", after removing full and partial duplicates, and removing hugs that involved animals. Inclusion criteria were: subjects are not carried or jumping (which excluded hugs between children and adults), they are facing each other (i.e., not sitting next to each other), their trunks are in firm contact, the couples are not kissing, and the posture is not sexually arousing. The complete analysis was performed three times independently, resulting in 398 unique photos which were rated for the side of the hugging (right or left trunk contact).

To obtain photos of kissing pairs, we performed the same procedure using the most efficient key words "airport kiss", "Lufthafen Kuss", "Begrüßungskuss", "Mutter Kuss", and "social kiss". We applied the same inclusion criteria as for the hug pictures, except that for kissing no tight trunk contact was present. Instead, the lips of both subjects were required to make contact with the partner's face. This procedure resulted again in 398 unique photos of kissing couples.

None of the tested subcategories (e.g., males, females, mixed couples, gray photos, outside, happy, sportive) gave substantially different (i.e., more than a few percent) results

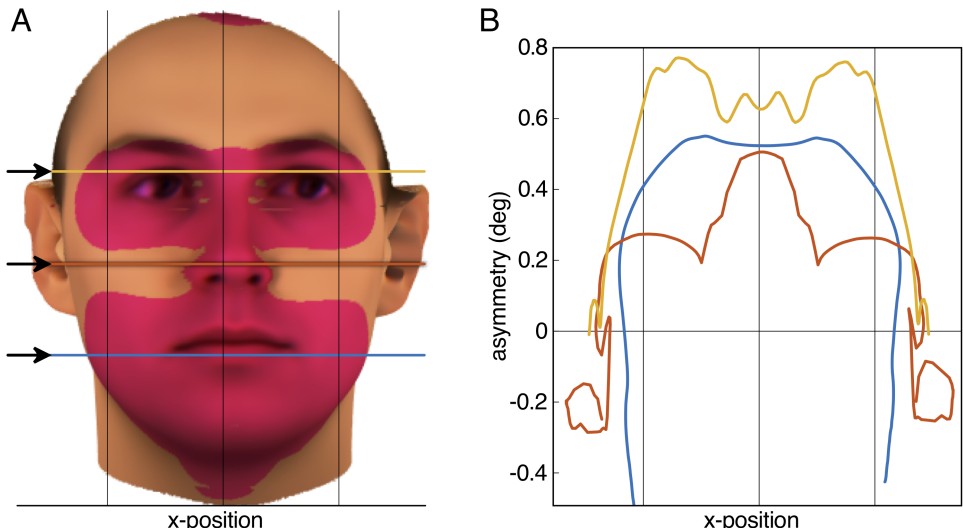

**Figure 2** **Result of the aurofacial asymmetry in adults.** (A) The average face from $N = 200$ (*Blanz & Vetter, 1999*; *Troje & Bülthoff, 1996*) with the tragus of the ears aligned with the horizontal axis of the image plane. Red-shaded area denotes significant aurofacial asymmetry to the left side. The asymmetry with respect to the symmetric reference face is displayed exaggerated by a factor five. (B) Magnitude of the aurofacial asymmetry (with respect to the symmetric reference face) along the three colored lines in (A). The asymmetry expresses the angular difference for each vortex between the average face and the symmetric reference face. Vertical lines in both panels are symmetrically arranged around the mid-plane between the tragi.

to the total, neither for hugging nor for kissing. A small number of photographs (less than 1%) were also found as mirror-symmetric versions. These were excluded from the analysis. The photographs cannot be published in order to protect their privacy.

## RESULTS

The aurofacial asymmetry in the adult database of *Blanz & Vetter (1999)* and *Troje & Bülthoff (1996)* is visualized in Fig. 2A. As predicted, the facial region, consisting of the eyes, nose and mouth was positioned significantly to the left with respect to the mid-plane between the tragus of the ears (red-shaded region: $p < 0.01$ Bonferroni-corrected, $p < 1.3 \cdot 10^{-7}$ uncorrected). This asymmetry was maximally 0.8° in the eyes (Fig. 2B). Table 1 shows the average asymmetry for each of the landmarks. Note that the standard deviations are more than twice as large as the average asymmetries (Cohen's statistical effect size $d = 0.44$).

The majority of adult faces from two photographic collections was found to be shifted to the left (60% left, 25% right, sign test: $p < 0.0001$), on average by 0.26° (s.d. $= 0.92$°, $t(112) = 3.0$, $p = 0.003$, Cohen's $d = 0.28$; Fig. S1).

The aurofacial asymmetry of preadolescent children was based on mean values of 3D facial landmarks from *Klingenberg et al. (2010)*. Table 2 presents the computed aurofacial asymmetries of four populations of pre-adolescent children (total $N = 168$, mean age 4.4–13.7 years), ranging from 0.81°–2.16°. According to a signed $t$-test, these mean values

**Table 1** Mean and standard deviation of the aurofacial asymmetry (in degrees) in the landmark positions of adult faces (same data as in Fig. 2).

| Landmark | Mean (s.d.) |
| --- | --- |
| frontotemporale | 0.35 (1.49) |
| exocanthion | 0.51 (1.36) |
| endocanthion | 0.69 (1.18) |
| nasion | 0.50 (1.26) |
| pronasion | 0.55 (1.33) |
| subnasion | 0.58 (1.30) |
| labiale superior | 0.58 (1.29) |
| menton | 0.44 (1.10) |
| average | 0.52 (1.19) |

**Table 2** Mean aurofacial asymmetries (degrees) in young and pre-adolescent children and adults. Cape: black children from Cape region, South Africa; Finnish: children from Finland; Control: healthy control group; Exposed: diagnosed with fetal alcohol syndrome (FAS). Children's data were reanalysed from *Klingenberg et al. (2010)* (see Methods). Adult data are the same as in Fig. 2.

| Population | Group | Age | (s.d.) | N | Angle |
| --- | --- | --- | --- | --- | --- |
| Cape | Control | 4.4 | (1.0) | 29 | 2.0 |
| Cape | Exposed | 5.2 | (1.3) | 49 | 2.2 |
| Finnish | Exposed | 13.2 | (3.6) | 40 | 0.8 |
| Finnish | Control | 13.7 | (3.6) | 50 | 0.9 |
| Adult | – | – | – | 200 | 0.52 |

are significantly different from zero ($t(3) = 4.1$, $p = 0.013$, Cohen's $d = 2.0$). Figure 3 shows the reduction of the asymmetry with the age. This reduction is consistent for all landmarks, but is more dramatic for the eyes' canthions than for the other landmarks.

The predictions on kissing and hugging were tested by analyzing photographic pictures taken from the internet. The results show, as predicted, that human kissing behavior is systematically biased to the left side of the face (68.1%, $N = 398$, sign test for binary data: $p < 0.00001$), while hugging was biased to the right side of the trunk (77.6%, $N = 398$, sign test: $p < 0.00001$). See also Table S1.

## DISCUSSION

All results were in line with the predictions. Moreover, the kissing bias agrees accurately with earlier findings (*Güntürkün, 2003* reported 65% on the basis of 124 pairs). Thus, the present study provides the first empirical validation of predicted effects for the Axial Twist Hypothesis (ATH). These include a new kind of asymmetry of the face (the *aurofacial asymmetry*), and the asymmetric hugging behavior. To our knowledge, these effects have never been reported before. These effects are not trivial, meaning that they are not easily explained with other theories that we know of. Since the other twist theory, the STM, is not a developmental model it does not predict systematic asymmetries. In the following we will first discuss anatomical effects, then behavioral ones and we will end with an outlook.

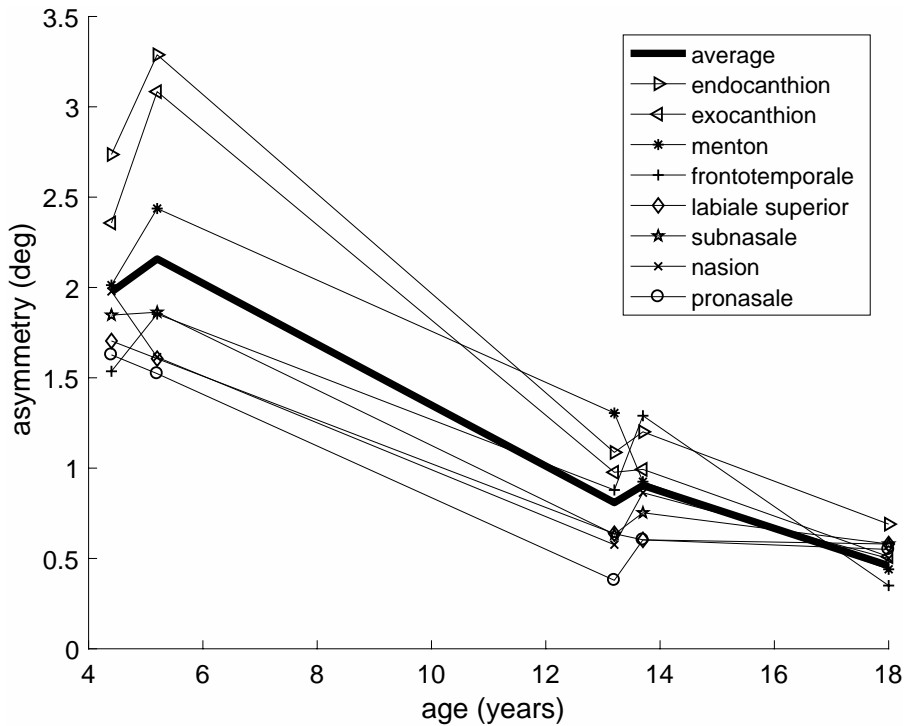

**Figure 3** Age dependence of the aurofacial asymmetry, for the same groups and landmarks as in Tables 1 and 2.

The predicted aurofacial asymmetry is a new kind of facial asymmetry. The ATH also accurately predicts further published anatomical asymmetries. The first one concerns the cerebral asymmetry. Functional and anatomical cerebral asymmetries are not specific for humans, but are widespread among vertebrates (*Rogers, Vallortigara & Andrew, 2013*). The cerebral frontal lobe is predicted to arrive to the left of the midline and the occipital lobe will arrive to the right of the midline; the right central sulcus and the right temporal lobe are more frontal than those on the left side. This rotational asymmetry (cf. Fig. 4A) is also known as Perisylvian asymmetry (*Geschwind & Levitsky, 1968*) or Yakovlevian torque (*LeMay, 1976*; *Toga & Thompson, 2003*). It is even reflected in the general pattern of cortical thickness (*Kong et al., 2018*) (Fig. 4A). The ATH predicts exactly this pattern, whereas we know of no alternative explanation for this pattern of cerebral asymmetries.

Since the spine is crucial for locomotion, it is predicted to be highly symmetrical for most of it length. However, since the ribcage provides much additional mechanical stability, it can be expected that the thoracic spine is, on average, less symmetric than the lumbar and cervical spine, even in normal healthy subjects. This is indeed the case. In healthy males T6 was found on average to be rotated by 2.4°, and in healthy females T7 was rotated, on average, by 3.1° (Fig. 4A; *Kouwenhoven et al., 2006*).

In idiopathic scoliosis this systematic rotation of the vertebrae is generally much larger than in healthy subjects, but always in the same direction (*Aaro & Dahlborn, 1981*; *Lee, Suk & Chung, 2004*). The etiology is still unclear (*Yagi, Machida & Asazuma, 2014*), but genetic
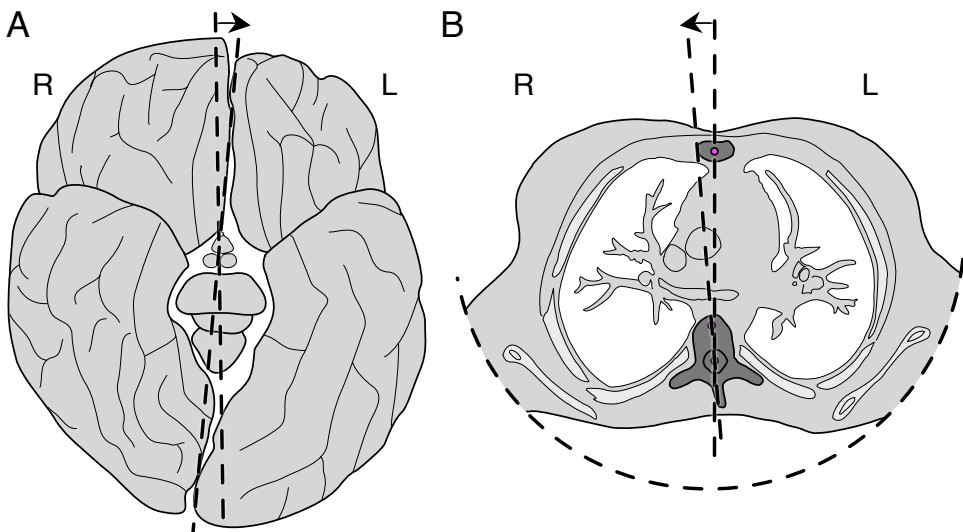

**Figure 4** **Opposite rotational asymmetries as viewed from below (in accordance with medical conventions, so that right [R] and left [L] are mirrored).** The dorsal/occipital side is at the bottom. (A) The Yakovlevian torque of the cerebrum (exaggerated). (B) The angular torsion of a t5 vertebra of a typical healthy subject. Note the opposite direction of the rotational asymmetry. Redrawn from *Kouwenhoven et al. (2006)* and *Toga & Thompson (2003)*. Source: M.H.E. de Lussanet, https://www.wikimedia, Creative Commons CC0 1.0.

factors related to bone growth have been shown to be involved (*Nowak, Szota & Mazurek, 2012*). The ATH provides a potential biomechanical mechanism for idiopathic scoliosis, because a strongly asymmetric orientation of vertebrate bodies will be biomechanically difficult to stabilize. This will be even more so in phases of rapid growth when the intervertebral muscles and fascia that grant biomechanical stability are strongly stretched and thus may for some period may not operate in their optimal self-stable length-range (*Wagner & Blickhan, 2003*; *Wagner et al., 2005*). In sum, these known asymmetries of the cerebrum and the thoracic spine, even in normal healthy humans, are as predicted by the ATH, if the axial twist underlies a control residual of the evolutionary selective mechanism.

We found that the aurofacial asymmetry in adults was highly significant, but still quite small. This is hardly surprising, since the symmetry of the human face can be expected to be under strong sexual selection. Indeed, we found that the average asymmetries of 0.4–0.7° are considerably smaller than the standard deviation of the adult population of the database that we used (cf. Table 1). Consequently, a considerable part of the population shows opposite asymmetries (cf. Fig. S1), so that in part of the population the selective pressure will be in the opposite direction, explaining the control error of selection for bilateral symmetry.

Since sexual selection does not act before puberty, we also hypothesized that the aurofacial asymmetry is larger in children and reduces with age. The gradual reduction of the aurofacial asymmetry is strong support for the ATH. The results also suggest that the asymmetry of the eyes might be slightly larger than that of the midline structures. Nevertheless, the asymmetries of the different landmarks are highly correlated (cf. Fig. S2),

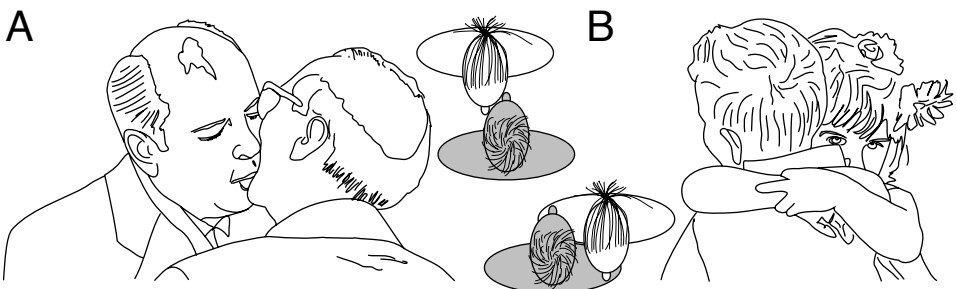

**Figure 5** **Examples of left kissing (A) and right hugging (B).** The two schemas show a top view of the opposite behavioral asymmetries. Source: M.H.E. de Lussanet, www.wikimedia, Creative Commons CC0 1.0.

so that they may indeed underly a single mechanism (i.e. the ATH). Finally, the data suggest that the aurofacial symmetry is very similar between children diagnosed with fetal alcohol syndrome (FAS) and healthy controls of the same age.

The findings of the embracing study confirm the hypothesis (Fig. 5B). They also accurately reproduce earlier observations on airports (*Turnbull, Stein & Lucas, 1995*; *Packheiser et al., 2019*) and findings of an experiment in which subjects embraced a doll (*Packheiser et al., 2019*).

The kissing results also confirm the hypothesis (Fig. 5A) and reproduces the findings of airport observations (*Güntürkün, 2003*), experiments with dolls (*Barrett, Greenwood & McCullagh, 2006*; *Ocklenburg & Güntürkün, 2009*; *Van der Kamp & Cañal-Bruland, 2011*), as well as with couples (*Barrett, Greenwood & McCullagh, 2006*) and questionnaires (*Chapelain et al., 2015*). Earlier studies have revealed clear regional cultural influences: for example in some French cities (*Chapelain et al., 2015*), as well as in native Palestinian and Jewish Israelis (*Shaki, 2013*), the kissing bias is reversed, whereas in a conservative muslim country (Bangladesh) the kissing bias is as in the other studies (*Karim et al., 2017*). Also, the bias in kissing and hugging behavior is strongly reduced by emotional contexts (*Packheiser et al., 2019*). For example, no bias was found in a public kiss between strangers (*Sedgewick, Holtslander & Elias, 2019*). Thirdly, the kissing bias can be influenced by a lateral head tilt. For example, when kissing a doll head that is either 5° tilted to the right or 15° to the left resulted in a bias of almost 100% to the left and right side of the face respectively (*Van der Kamp & Cañal-Bruland, 2011*). Finally, both the kissing and hugging bias seem to be reduced in left handers (*Ocklenburg & Güntürkün, 2009*; *Karim et al., 2017*, although this was not so in *Van der Kamp & Cañal-Bruland, 2011*).

According to earlier proposals, the hugging bias results from right-hand dominance (*Ocklenburg et al., 2018*). Accordingly, the right hand leads by grabbing across the partner's shoulder and so explains the correlation of the bias with handedness. A closer view on the act of embracing makes this explanation unlikely: when two opposites use their right hand to pull the left shoulder, the left sides of the trunks will tend to touch, which is the opposite of the observed bias.

It has been proposed that the kissing bias is caused by a presumed tendency to right-turn the head, which might have its roots in the development of handedness (e.g. *Güntürkün, 2003*; *Ocklenburg et al., 2018*). An asymmetric kiss can be conducted by a head turn, but also by a lateral shift of the body, as well as by a side-tilt of the head without a turn. Although it is difficult to tell from the photos of our dataset which of the three mechanisms caused the asymmetric kissing. However, in the analysed photos a side tilt is very common, as well as a lateral shift of the body. Only few photos indicate that a side turn of the head led to the asymmetric kissing. It has been shown that a side tilt of the doll face can indeed cause a kissing bias (*Van der Kamp & Cañal-Bruland, 2011* see above).

According to the present hypothesis the kissing asymmetry is not due to a tendency to move asymmetrically, but rather to a tendency to aim with the side of the face. The present hypothesis is therefore neutral as to how the asymmetric final kissing posture is reached, whether by a tilt, a turn or a lateral shift. Thus, the present results cannot clearly differentiate between the theories that predict a kissing bias. A systematic experiment may shed more light on the plausibility of the different explanations.

We thus showed that humans also *behave* as twisted creatures, as predicted by the ATH. Asking people why they kiss or hug this way, or to try it the other way leads to responses such as "it somehow feels better, more natural like this." We thus tend to kiss as if the ventral side of the face has not quite arrived in the centre, but is still located to the left. Correspondingly, we tend to hug as if the ventral trunk is located to the right of the sagittal plane.

Systematic behavioral asymmetries are present in newborns and develop in the first years of life (*Michel, Sheu & Brumley, 2002*). Even if the initial biases are small, they can be expected to lead to a considerable behavioral bias because motor learning processes will strengthen successful movements. Consequently, a small behavioral bias will tend to produce slight lateralised biases in the number of successful movements. Given that asymmetric behavioral biases tend to be exaggerated during early childhood (*Nelson, Campbell & Michel, 2013*), we predicted that the anatomical asymmetries should be reflected in behavioral asymmetries in the trunk (hugging) and the face (kissing).

Similar to our reasoning, *Güntürkün (2003)* argued that the kissing bias is retained from new-born behavioral asymmetry (*Coryell & Michel, 1978*; *Konishi et al., 1987*). He argued that the bias of new-borns to turn the head to the right when in supine position (*Konishi, Mikawa & Suzuki, 1986*; *Konishi et al., 1987*; *Ververs et al., 1994*) explains the kissing bias. The result for the asymmetric kissing behavior taken for itself can thus be explained by (at least) two alternative hypotheses. However, the combination of a left-sided bias for kissing and a right-sided bias for hugging is only predicted by the ATH, so the behavioral results, too, give strong evidence in favor of the ATH.

As mentioned above, a similar twist hypothesis has been developed independently. Like the ATH, the Somatic Twist Model (STM), was designed to explain the contralateral organization of the forebrain (*Kinsbourne, 1978*; *Kinsbourne, 2013*). It adapted the Dorsoventral Inversion Hypothesis (DIH) (*Geoffroy-Saint-Hilaire, 1822*; *Nübler-Jung & Arendt, 1994*; *Lowe et al., 2003*), by proposing that the body, but not the anterior head region, is inverted dorsoventrally as compared to protostome animals such as arthropods

and annelids. The STM does not tell why the body, but not the anterior head, might be inverted nor does it provide a developmental mechanism.

A dorsoventral inversion is known from cephalochordates, the adults of which bury in the sand with the dorsal side turned downward. Some starfish (Echinodermata) have, at the end of development for a short time their mouth turned upward while their aboral side is connected by a small stalk to the substrate. This stalk is then detached and the animal turns upside down so that the mouth is turned down (*Atwood, 1973*; *Gemmill, 1912*; *Gemmill, 1914*; *Gemmill, 1920*). A dorsoventral inversion following asymmetric development is thus not uncommon in deuterostomes.

Our results have important implications for the understanding of the relation between asymmetries found in vertebrates and those in other deuterostomia. The classic example of deuterostome asymmetric development is the lancelet (Cephalochordata), whose mouth migrates from the left side to the center, after which the symmetric adult animal buries itself in a dorsoventrally inverted orientation with the mouth up. The urochordates (tunicates), which are more closely related to vertebrates tend to settle as sessile, sac-like animals, with their mouth twisted towards an upward direction for optimal filter feeding. The third example of strongly twisted deuterostomes are the echinoderms, in which the mouth and anal regions migrate in opposite directions in a very complex developmental process of twisting deformations (*De Lussanet, 2011*). Thus, the development of anatomical twists by complex asymmetric developmental processes seems to be a hallmark of deuterostomes.

In contrast to the STM, the ATH is not an adaptation of the dorsoventral inversion hypothesis. However, *De Lussanet & Osse (2012)* discussed the possibility that a dorsoventral inversion followed the evolution of the axial twist. As the vertebrate mouth appears very late in embryological development, it has been impossible, before, to show if the mouth twists with the ears or with the face and therefore it was, at the time, impossible to decide if the ATH can explain the dorsoventral inversion (*De Lussanet & Osse, 2012*). Given that the current results suggest that the asymmetry of the mouth is consistent with the eyes and the nose, the current results strongly suggest that the mouth belongs to the forehead region from a developmental perspective.

According to the present broad consensus, vertebrates, including ourselves, are bilaterally symmetric members of the so called bilaterians. The well-known asymmetries of the heart and other internal organs have been regarded as specific morphological peculiarities and as minor exceptions to the general symmetric pattern. The present results challenge this traditional view and have far reaching consequences for our understanding of development, morphology and behavior. According to this new view, the symmetry of the vertebrate body is only superficial and a secondary adaptation to the active mode of life of vertebrates (as compared to the many sessile forms of tunicate relatives). The heart and internal organs retain their original orientation in the body whereas the other body parts only assume a bilateral symmetry by virtue of highly advanced and complex developmental processes.

## CONCLUSIONS

On the basis of the Axial Twist Hypothesis (ATH) for the evolution and development of the contralateral forebrain and the optic chiasm we made five specific predictions regarding anatomical and behavioral asymmetries. Two of these could be confirmed on the basis of existing literature, whereas three represent new and non-trivial findings. The predictions were tested on the basis of human data.

1. A new kind of facial asymmetry, the systematic left-sided aurofacial asymmetry, was predicted and confirmed.
2. As predicted, this aurofacial asymmetry was larger in children's faces than in adult ones. It was not different for children with fetal alcohol syndrome of the same age.
3. As confirmed by literature, the Yakovlevian torque and the rotational asymmetry of the thoracic vertebrae in healthy human populations are in opposite directions and these are as predicted by the ATH.
4. As confirmed by literature, humans tend to kiss with the left side of their face, as predicted by the ATH. This finding was accurately reproduced by the current findings.
5. As predicted by the ATH, we found that humans tend to hug oppositely to kissing, using the right side of the trunk.
6. Although some of the above findings might be explained by alternative hypotheses, we are not aware of any other theory that can explain this complex and counterintuitive pattern of results.
7. With respect to the evolution of the twist, we have provided the first empirical evidence that the mouth turns in the same direction as the rostral head. This provides a clear evolutionary perspective and links the ATH with the dorsoventral inversion hypothesis.
8. By this, the ATH becomes the only theory of the contralateral forebrain and the optic chiasm to have undergone empirical testing. These five specific predictions add to the already demonstrated explanatory power which reaches far beyond the phenomena for which the theory was originally proposed.

## ACKNOWLEDGEMENTS

I am indebted to Prof. Dr. Niko Troje for providing an initial analysis script of the 3D facial analyses of the adult data and for helpful discussions and comments. I thank Laurens de Lussanet for his help in the analysis of kissing and hugging. Niklas Lohmann, Prof. Dr. Martin Fischer, Dr. Nils Schlüter, Prof. Dr. Heiko Wagner, Dr. Kim Boström, and Dr. Christian Puta for fruitful discussions and comments. I thank the reviewers and the editor for their constructive comments.

### Funding

The author received no funding for this work.

### Competing Interests

The author declares that there are no competing interests.

## Author Contributions

- Marc H.E. de Lussanet conceived and designed the experiments, performed the experiments, analyzed the data, contributed reagents/materials/analysis tools, prepared figures and/or tables, authored or reviewed drafts of the paper, approved the final draft.

## Ethics

The following information was supplied relating to ethical approvals (i.e., approving body and any reference numbers):

The current work was approved by the ethics committee of FB07 of the University of Münster (Ethical Application Reference EK-17-25).

## Data Availability

The Matlab code for the analysis of 3D surface data, script, and datasets are available in FigShare: de Lussanet, Marc H.E. (2019): Matlab script for computing the aurofacial asymmetry. figshare. Software. https://doi.org/10.6084/m9.figshare.8050496.v1.

The code was written to analyze 3-D images taken from http://faces.kyb.tuebingen.mpg.de (which requires account registration to access).

Supplementary results of the aurofacial analyses are available in Figs. S1 and S2.

The analysis of the aurofacial asymmetry in the frontal face photos is available in Fig. S3. This Figure shows all the analysed photos and the Table with the aurofacial asymmetries.

The raw results of the kissing and hugging analysis are available in Table S1. Note that the photos cannot be published (to protect the privacy of the subjects) but they can be provided by contacting the author if an ethical board approval statement is provided.

The images from the internet search used for the kissing and hugging analysis cannot be made available due to privacy issues.

## Supplemental Information

Supplemental information for this article can be found online at http://dx.doi.org/10.7717/peerj.7096#supplemental-information.

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
