# Peer review of "Opposite asymmetries of face and trunk and of kissing and hugging, as predicted by the axial twist hypothesis"

_PeerJ, doi:10.7717/peerj.7096_

## Round 0.1 · original submission · Major Revisions

The Reviewers are in general favourable. Please check their comments and provide proper amendments or rebuttal. On my side, I would recommend the author to consider also the large literature on brain asymmetry in vertebrates that is somewhat related to his hypothesis, in particular see Chapter 3 on the origins of asymmetry in Chordates in Rogers et al (2013). Divided Brains. Cambridge University Press, and for mouth asymmetry:
https://www.scientificamerican.com/article/evolutionary-origins-of-your-right-and-left-brain/
I am looking forward to receiving the revised version of this interesting paper.
Best regards,
gv

Reviewer 1 ·

Basic reporting

- The manuscript is written in clear, unambiguous, professional English language throughout.
- The Intro shows the context, but the literature is not well referenced and there are several highly relevant papers missing (see specific comment). This should be easily remedied, though.
- The structure follows standards in the field.
- The figures are great.
- I could not find any reference in the paper whether or not the raw data are supplied. This should be mentioned somewhere in the text. The raw data itself is provided as an Excel Sheet on the homepage.

Experimental design

- The author provides original primary research within Scope of the journal.
- The research question is well defined, relevant & meaningful.
- It is stated how the research fills an identified knowledge gap.
- The standard of the investigation seems to be high. Since no participants are tested but databases and pictures are analyzed, no ethical approval was needed it seems
- The Methods are described with sufficient detail & information to replicate. I must admit I am not an expert for analysis of 3D surface data, so I cannot judge this part of the methods 100%

Validity of the findings

No comments.

Additional comments

General comments:
Intro:
- Page 2, line 91: The statement “Since the evolutionary selection for bilateral symmetry is due to locomotion and sexual selection, it will only affect the external shape of the body.” Needs citations.
- The section on asymmetries in kissing and hugging at the end of the introduction needs to be redone for several reasons. First the author omits pretty much all of the relevant literature by citing only one paper on kissing and none on hugging.
- For asymmetries in hugging please integrate the relevant literature:
- Turnbull, O.H., Stein, L., Lucas, M.D., 1995. Lateral Preferences in Adult Embracing: A Test of the “Hemispheric Asymmetry” Theory of Infant Cradling. The Journal of Genetic Psychology 156, 17–21. 10.1080/00221325.1995.9914802.
- Packheiser, J., Rook, N., Dursun, Z., Mesenhöller, J., Wenglorz, A., Güntürkün, O., Ocklenburg, S., 2018. Embracing your emotions: Affective state impacts lateralisation of human embraces. Psychological research. 10.1007/s00426-018-0985-8..
- Ocklenburg S, Packheiser J, Schmitz J, Rook N, Güntürkün O, Peterburs J, Grimshaw GM. Hugs and kisses - The role of motor preferences and emotional lateralization for hemispheric asymmetries in human social touch. Neurosci Biobehav Rev. 2018 Dec;95:353-360. doi: 10.1016/j.neubiorev.2018.10.007
- These papers are generally in line with the authors finding of a rightward bias for hugging.
- For kissing the authors state: “A left-sided bias of 65% has indeed been found for kissing (Güntürkün, 2003).” But if you read that paper the abstract reads “Here I show that twice as many adults turn their heads to the right as to the left when kissing, indicating that this head-motor bias persists into adulthood.” This is very confusing and the author need to rephrase or explain better why they make this assumption.
- Also in general pretty much all of the literature shows a rightward bias for kissing but is not cited. The following paper should not be ignored but integrated in the intro:
- Barrett, D., Greenwood, J.G., McCullagh, J.F., 2006. Kissing laterality and handedness. Laterality 11, 573–579. 10.1080/13576500600886614.
- Chapelain, A., Pimbert, P., Aube, L., Perrocheau, O., Debunne, G., Bellido, A., Blois-Heulin, C., 2015. Can Population-Level Laterality Stem from Social Pressures? Evidence from Cheek Kissing in Humans. PloS one 10, e0124477. 10.1371/journal.pone.0124477
- Karim, A.K.M.R., Proulx, M.J., Sousa, A.A. de, Karmaker, C., Rahman, A., Karim, F., Nigar, N., 2017. The right way to kiss: Directionality bias in head-turning during kissing. Scientific reports 7, 5398. 10.1038/s41598-017-04942-9
- Ocklenburg, S., Güntürkün, O., 2009. Head-turning asymmetries during kissing and their association with lateral preference. Laterality 14, 79–85. 10.1080/13576500802243689.
- Based on these papers the author should consider rephrasing the kissing hypothesis.

Results:
- Please provide effect size measures.


Discussion:
- The authors states: “All results were in line with the predictions. Moreover, the kissing bias agrees accurately with earlier findings (G¨unt¨urk¨un, 2003, reported 65% on the basis of 124 pairs).” This is not correct. As reported above, Güntürkün finds a right bias. Please rephrase. Also include the papers on hugging and kissing asymmetry mentioned above into the discussion.

·

Basic reporting

This manuscript is written clearly and well. I suggest, however, that figure be added to show the locations of the exocathion, endocanthion, nasion, menton, etc. in order to assist the more general reader.

Experimental design

This looks fine.

Validity of the findings

The author has provided empirical evidence supporting the Axial Twist Hypothesis.
As far as I can tell, the data are reliable and the conclusions are supported. Although the magnitude of the biases away from symmetry are small, the authors explain why this would be so and provide evidence for the predicted differences in magnitude of bias between adults and children. They also provide behavioural data (kissing and hugging) consistent with the direction of bias in the aurofacial region versus the body.

Additional comments

I enjoyed reading this paper. It adds evidence in support of a clearly presented hypothesis about development of structural asymmetry of the face versus body, and makes predictions about further measures of asymmetry.

---

## Round 0.2 · accepted · Accept

The Reviewers are happy with the revision, and after reading it I also concur with their judgements. Please note that Reviewer 2 noted that one reference was reported incorrectly (530: The citation of Rogers et al 2013 is incorrect. The title of the book is ‘Divided Brains: The Biology and Behaviour of Brain Asymmetries’). This can be fixed with proof correction (or directly at the Production stage).

Reviewer 1 ·

Basic reporting

Everything is fine in the revised edition.

Experimental design

Everything is fine in the revised edition.

Validity of the findings

Everything is fine in the revised edition.

Additional comments

The author did a good job in integrating my comments and give a more balanced view of the kissing/hugging asymmetry literature. I think the paper is now ready for acceptance.

·

Basic reporting

The paper has been improved by the changes made and, in my opinion, it is ready for publication. I have, however, found one more correction that needs to be made:

530: The citation of Rogers et al 2013 is incorrect. The title of the book is ‘Divided Brains: The Biology and Behaviour of Brain Asymmetries’.

Experimental design

Please see above.

Validity of the findings

Please see above.